# The Impact of Data Science Solutions on the Company Turnover

**Marian Pompiliu Cristescu** [1] [ID], **Dumitru Alexandru Mara** [1,*] [ID], **Lia Cornelia Culda** [1], **Raluca Andreea Nerișanu** [1] [ID], **Adela Bâra** [2] [ID] **and Simona-Vasilica Oprea** [2] [ID]

[1] Faculty of Economic Sciences, Lucian Blaga University of Sibiu, 550324 Sibiu, Romania; marian.cristescu@ulbsibiu.ro (M.P.C.); cornelia.culda@ulbsibiu.ro (L.C.C.); raluca.andreea.nerisanu@gmail.com (R.A.N.)

[2] Department of Economic Informatics and Cybernetics, Bucharest University of Economic Studies, 010374 Bucharest, Romania; bara.adela@ie.ase.ro (A.B.); simona.oprea@csie.ase.com (S.-V.O.)

* Correspondence: dumitrualexandru.mara@ulbsibiu.ro

**Abstract:** This study explores the potential of data science software solutions like Customer Relationship Management Software (CRM) for increasing the revenue generation of businesses. We focused on those businesses in the accommodation and food service sector across the European Union (EU). The investigation is contextualized within the rising trend of data-driven decision-making, examining the potential correlation between data science applications and business revenues. By employing a comprehensive evaluation of Eurostat datasets from 2014 to 2021, we used both univariate and multivariate analyses, assessing the percentage of companies that have e-commerce sales across the EU countries, focusing on the usage of big data analytics from any source and the use of CRM tools for marketing purposes or other activities. Big data utilization showed a clear, positive relationship with enhanced e-commerce sales. However, CRM tools exhibited a dualistic impact: while their use in marketing showed no significant effect on sales, their application in non-marketing functions had negative effects on sales. These findings underscore the potential role of CRM and data science solutions in enhancing business performance in the EU's accommodation and food service industry.

**Keywords:** big data analytics; revenue generation; Customer Relationship Management (CRM)

## 1. Introduction

In today's digital age, data have emerged as crucial assets for businesses, driving decision-making, innovation, and growth. The application of data science solutions, which involves extracting meaningful insights from vast amounts of data, has shown a significant impact on company revenues.

Our primary research question seeks to understand "To what extent do contemporary technological tools, specifically big data and CRM, influence e-commerce sales across different countries?".

Data are valuable resources that can leverage business partnerships, vertical integration, or diversification. Their study revealed that consistency, completeness, and protection of data, combined with a data-driven company profile, can lead to improved customer management and operational efficiency. Such enhancements directly contribute to better business performance and, consequently, increased revenues [1].

By establishing a direct link between the outcomes of data science methodologies and value drivers for customers, businesses can systematically derive value from insights generated by data science. This systematic value creation translates into enhanced customer experiences, leading to higher sales and revenue [2].

According to [3], the spread of real-time data across companies, given the availability of appropriate analytical tools and methods, can have a significant impact on the entire company. This suggests that real-time data analytics, a subset of data science solutions, can

provide businesses with timely insights, enabling them to make agile decisions that can boost revenues.

The contemporary landscape of e-commerce has seen a significant shift due to the integration of big data and Customer Relationship Management (CRM). A study by [4] reveals the profound implications of this merger, suggesting that businesses adopting this strategy become notably more aggressive in their marketing endeavors, leveraging tactics like push notifications to reach potential target audiences. This is further reinforced by [5], who identified shortcomings in traditional decision-making systems, highlighting the pivotal role of big data in bolstering both strategic and operational decisions, which in turn boosts marketing performance.

Drawing attention to the interplay between big data and CRM, [6] introduced a theoretical model delineating their combined influence on strategic sales performance. However, the relationship between these technologies and sales is not one-dimensional. The research of [7] underscores the role of regional differences, implying that the ramifications of big data on sales growth and gross margins might be mediated by region-specific factors. Ref. [8] delves into the B2B domain, affirming that big data analytics tied with customer relationship strategies can be a potent catalyst for enhanced sales growth.

The need to understand this intricate relationship between big data, CRM, and e-commerce sales has never been more pressing. The rationale is twofold: firstly, the digital transformation wave sweeping across industries, and secondly, the meteoric rise of e-commerce. As businesses chart their course in this digital era, understanding the drivers behind sales is pivotal. It is not merely about adopting technology; it is about leveraging it strategically. The digital arena of today boasts data of unprecedented volume, variety, and velocity, making the current juncture pivotal for businesses to comprehend and harness the potential of these technological marvels.

Despite a plethora of research, a discernible knowledge gap persists. While many studies like that of [4,5] have shone light on specific aspects of the interplay between big data and CRM, a holistic understanding of their synergistic influence on e-commerce sales remains elusive. This gap is further accentuated when we consider nuanced application strategies across different contexts and regions. Key literary works, such as the ones by [9,10], have contributed substantially to our understanding of big data's impact on domains like supply chain management and CRM, respectively. However, there's a call for a deeper dive. The work of [11] underscores the ambiguity surrounding the nuanced role of CRM in e-commerce, while the authors of [10] spotlight the broader implications of big data on e-commerce sales. This is further complicated by insights from [12], who warn of potential pitfalls tied to an over-reliance on big data analytics for e-vendors.

As businesses navigate the intertwined realms of big data, CRM, and e-commerce, it is imperative to bridge existing knowledge gaps. Only through a comprehensive understanding of these interdependencies can businesses truly unlock the vast potential these technologies hold.

## 2. Literature Review

Customer relationship management has evolved as an important tool for businesses in the digital age and in their electronic commerce activities. At its core, CRM is not just about managing interactions with customers but is a comprehensive approach to understanding, targeting, and building relationships with customers to drive business growth.

In recent years, there has been a rise in research evaluating the perceived business advantages of e-commerce. This study found, however, that the results vary substantially based on the differences between countries, the specific characteristics of businesses, and the types of businesses conducted. Ref. [13] highlights the benefits of e-commerce for small and medium-sized enterprises, in particular, market expansion and cost reduction.

Chaffey categorized the benefits of e-commerce in this context as tangible benefits and intangible benefits. Intangible benefits are difficult to identify and quantify, whereas tangible benefits are associated with quantifiable factors such as increased sales, decreased

costs, and market expansion. They are, however, closely related to tangible factors. For instance, a company that can shorten its product's life cycle can generate tangible benefits such as reduced costs, greater customer satisfaction, and, ultimately, a rise in sales [14].

Rather than cost reduction, the most significant advantages of electronic commerce are related to the competitive environment. Increased sales and competitiveness are additional significant advantages. In addition, companies emphasize the significance of growing their customer base and improving customer service. Ref. [15] emphasizes the effectiveness of online media during the COVID-19 pandemic in boosting product sales.

According to [16], one of the most significant benefits for small businesses is the increased ability to obtain information about customers and suppliers. Additionally, they can benefit from operating across regional and national borders. Because technology is global, companies can expand their market presence by entering global markets. Ref. [17] also emphasizes the importance of e-commerce in international business and its implications for the study of international business.

Nonetheless, through electronic commerce, small businesses can increase their ability to communicate with consumers, suppliers, and competitors to the same extent as many of the world's largest corporations, thereby enhancing their competitiveness. Ref. [18] concurs with this result. They suggest that, due to their size, small businesses will be more adaptable to changing conditions and will benefit from the increased speed and adaptability that electronic commerce provides.

In the realm of e-commerce, CRM takes on an even more significant role. E-commerce businesses operate in a highly competitive environment where customer preferences can shift rapidly, and the market dynamics can change overnight. In such a scenario, having a robust CRM system can be the difference between success and failure.

Ref. [19] emphasized the importance of CRM in electronic commerce, suggesting that it is a concept designed to increase companies' profitability by enabling them to identify and concentrate on their profitable customers. More specifically, the application of CRM in electronic commerce, often referred to as Electronic Customer Relationship Management (e-CRM), is a critical success factor in the digital marketplace. This is especially true for business-to-consumer companies and smaller companies that might not have the vast resources of larger corporations but still need to effectively manage their customer relationships to thrive in the online space. The authors argue that the duration a company has been on the Web does not necessarily correlate with its success in electronic commerce; instead, the effective application of CRM strategies plays a more crucial role [19].

One of the primary advantages of e-CRM is its ability to provide personalized and customized support to customers. Ref. [20] highlights the strategic role of e-CRM in offering tailored solutions and experiences to customers. This personalization is crucial in the e-commerce landscape, where customers are often inundated with choices. By understanding customer preferences, behaviors, and purchase histories, e-CRM systems can deliver targeted offers, product recommendations, and content, thereby enhancing the overall shopping experience and increasing the likelihood of conversions [20,21] emphasized the importance of enhancing "user experience" to foster loyalty and improve returns. As businesses seek to adapt to these changes, the integration of data science solutions, like CRM, becomes invaluable. CRM systems offer insights into user behavior, enabling businesses to tailor user experiences in applications. Such personalized interactions can lead to increased conversions and turnover. This is especially pertinent when evaluating user expectations for specific functionalities, such as mobile e-commerce application features.

However, while the potential benefits of e-CRM are evident, its implementation in the retail sector has been varied. A study by [22] analyzed the availability of e-CRM features on retail websites and their relationship with consumer satisfaction and site traffic. Their findings indicated that while e-CRM features can significantly enhance consumer satisfaction, many standard retailers lag behind in their implementation. This gap presents both a challenge and an opportunity. Retailers who can effectively integrate e-CRM features

into their online platforms stand to gain a competitive edge, while those who neglect this aspect risk alienating their customer base.

The integration of customer relationship management into a company's strategy has profound implications for its marketing performance. In the e-commerce landscape, where competition is fierce and customer loyalty is paramount, the role of CRM becomes even more critical.

Ref. [23] conducted a study that delved into the effects of CRM on marketing performance. Their findings were unequivocal: CRM significantly influences marketing performance. The study revealed a strong correlation, with a path coefficient of 0.79, indicating that companies that effectively implement and manage their CRM strategies are more likely to see enhanced marketing outcomes.

However, the impact of CRM is not limited to marketing performance alone. Another study by [24] explored the relationship between CRM, innovation, and performance advantages. Their research demonstrated that developing close relationships with customers enhances a firm's ability to innovate. This innovation, in turn, has a direct positive impact on performance. The study supports the idea that CRM is not just a tool for managing customer interactions but can be a catalyst for broader organizational improvements, driving both innovation and performance.

Also, a study by [25] explored the impact of e-CRM on service quality, particularly in private hospitals in Jordan. The results demonstrated that electronic customer relationship management had a positive impact on service quality. This finding underscores the importance of e-CRM not just in retail e-commerce but also in sectors like healthcare, where service quality can significantly influence customer trust and satisfaction.

Another dimension of service management in e-CRM is how businesses handle service failures and customer complaints. A study by [26] emphasized that successful service management is at the core of e-CRM. Their research on the impact of e-service failures and customer complaints on e-CRM revealed that how businesses address and rectify these issues plays a crucial role in shaping customer perceptions and loyalty. Effective complaint resolution can not only mitigate the negative effects of service failures but also enhance customer trust and loyalty in the long run.

The capabilities of CRM systems have expanded significantly with the advent of new technologies, especially Big Data Analytics (BDA). These capabilities not only enhance the core functions of CRM but also serve as mediators that amplify its impact on sales performance.

With the rise in online interactions and data-driven decision-making, there's an observable surge in the volume, velocity, and diversity of data from various sources about consumers, processes, or environments. Ref. [27] highlights that breakthroughs in areas like machine learning and big data have accelerated advancements in predictive analytics. By integrating big data and machine learning, organizations have the capability to build and employ models in their data repositories to predict outcomes, gauge potential risks, or make analytical forecasts. Ref. [28] stressed the importance of extracting only the data pertinent to decision-making from the vast amounts of available data, relegating the rest to different applications, or disregarding it altogether. Ref. [29] notes the fast-paced evolution of big data analytics, a tool that bridges numerous businesses and institutions, enabling them not only to benefit from the data but also to enhance customer engagement, a sentiment echoed by [30].

Ref. [31] delved into the relationship between big data analytics, CRM capabilities, and perceived sales performance. Their findings indicated that BDA and CRM capabilities share a strong positive impact on perceived sales performance. This suggests that the integration of BDA into CRM systems can provide businesses with deeper insights into customer behaviors, preferences, and trends, thereby enabling them to tailor their marketing strategies more effectively and drive sales.

Another dimension of CRM capabilities is the integration of the Internet and database marketing. A study by [32] highlighted that such integration enhances the effectiveness of

CRM practices. By leveraging the vast amounts of data available online and combining it with traditional database marketing techniques, businesses can create more targeted and personalized marketing campaigns. This not only improves customer engagement but also leads to better conversion rates and increased sales.

In the dynamic world of e-commerce, where businesses vie for customer attention and loyalty, CRM software has emerged as a game-changer. This chapter delves into the profound influence of CRM software on e-commerce sales, drawing from empirical research and insights.

Ref. [33] highlighted that differences in CRM effectiveness lead to significant variations in sales processes. Their research emphasized that the geography of firms does not qualify these influences, suggesting that the benefits of CRM are universal across different regions and markets.

A study by [34] supported the hypothesized influences of customer orientation, customer-centric organizational systems, and CRM technology on CRM capabilities. Furthermore, they found that CRM capabilities directly influence organizational performance, which includes sales outcomes.

Ref. [35] emphasized the role of e-CRM in enhancing customer satisfaction in online shopping. Their study provides insights for managers and marketers to implement e-CRM effectively, aligning it with the current needs and requirements of consumers. Such alignment can lead to increased customer loyalty and, consequently, higher sales.

Additionally, [11] provided empirical evidence that the adoption and utilization of CRM positively impact sales performance, sales effectiveness, and collaboration. This suggests that businesses that effectively integrate CRM into their operations can expect enhanced sales outcomes.

## 3. Materials and Methods

We utilized datasets from Eurostat, focusing on the accommodation and food service industry within the European Union. The datasets, classified under the NACE Rev.2 activity, offered a harmonized classification system, allowing for a standardized and comparative analysis across EU countries. The first dataset, "Big Data Analysis by NACE Rev.2 activity [ISOC_EB_BDN2__custom_7625275]", provided insights into the percentage of companies employing big data analytics, incorporating both internal and external data sources until the year 2020. The second dataset, "E-commerce Sales of Enterprises by NACE Rev.2 activity [ISOC_EC_ESELN2__custom_7176858]", delineated the prevalence of e-commerce sales among these companies. The third and fourth datasets, both titled "Integration of Internal Processes by NACE Rev.2 activity [ISOC_EB_IIPN2__custom_6978957]", illustrated the extent to which enterprises deploy Customer Relationship Management (CRM) tools for marketing analytics and internal information sharing about clients. By amalgamating and analyzing these datasets, we aimed to uncover intricate patterns and correlations, offering a panoramic view of the digital transformation landscape within the accommodation and food service industry across the European Union. As it can be seen in Figure 1, we have used three independent variables and one dependent variable to investigate the influence of data science solutions, such as customer relationship management tools, on the capability of the company to make e-commerce sales.

The dependent variable, the percentage of enterprises engaging in e-commerce sales, was selected as an operationalized metric for company turnover. The assumption underlying this choice is that e-commerce sales constitute a significant and measurable component of total sales, directly influencing turnover. This variable allows for a nuanced exploration of the financial impact resulting from companies' strategic shift towards online sales platforms.

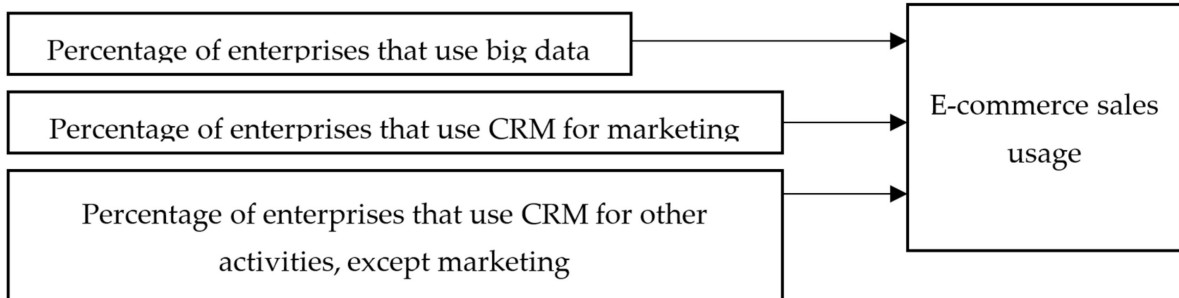

**Figure 1.** Dependent and independent variables.

The independent variables, including the percentage of enterprises using big data and CRM software, were identified based on their potential to directly and indirectly augment company turnover. Big data analytics is instrumental in extracting actionable insights, enhancing operational efficiency, and personalizing customer experiences, thereby potentially boosting sales and turnover. Similarly, CRM software, used for marketing and other activities, plays a pivotal role in customer engagement, retention, and loyalty, factors that are intrinsically linked to sales volume and, by extension, turnover.

These variables were measured in percentages at the national level, offering a standardized metric for cross-comparative analysis over a dataset that spans 28 countries and 4 years, resulting in 112 data points for each variable. The uniform methodology ensures that findings are not only indicative of temporal trends but also offer insights into geographical and national variations in the impact of data science solutions on turnover.

The selection was, therefore, anchored in the need to empirically evaluate the symbiotic relationship between internal technological adoptions—big data and CRM—and their resultant external financial performance impact, encapsulated by the e-commerce sales, within the thematic confines of data science solutions' influence on turnover.

We begin by exposing some descriptive statistics, including mean, maximum, and minimum values for each analyzed variable. After the descriptive statistics, visual maps were included in order to facilitate the visualization of big data, CRM, and e-commerce usage at the EU level in 2014 and 2021.

In order to argue our main hypothesis, we have performed three Student's T-tests to observe if there is a significant difference between e-commerce country-level usage when big data and CRM usage is below and above 20%. For the selection of a proper T-test, we first performed an F-test for the equality of variances.

After testing the hypothesis, we performed a multivariate linear regression. In order to ensure the reliability of the regression, we first tested the series for autocorrelation (by using Durbin–Watson procedure) and stationarity (Dickey–Fuller test). After the coefficients of the regression were estimated, the goodness of fit was analyzed. Equation (1) was used for the linear regression. Also, a 3-dimensional plot was used to enhance visualization of the relationship among the analyzed variables.

$$E\_commerce\ sales = c + \beta_1 * Big\ data + \beta_2 * CRM_{marketing} + \beta_3 * CRM_{other\ activities} \quad (1)$$

- *E-commerce sales*: The dependent variable indicates the proportion of companies involved in online sales.
- $c$ = The baseline level of e-commerce sales when all independent variables are zero.
- $\beta_1 * Big\ data$. Coefficient: Reflects the impact of big data analytics on e-commerce sales. Variable: The percentage of companies analyzing big data from any source.
- $\beta_2 * CRM_{marketing}$ Coefficient: Measures the influence of CRM for marketing on e-commerce sales. Variable: The percentage of companies employing CRM tools for marketing.

- $\beta_3 * CRM_{other\ activities}$ Coefficient: Reflects the effect of CRM's non-marketing functions on e-commerce sales. Variable: The percentage of companies using CRM for activities other than marketing.

Software used in statistical processing were IBM SPSS Statistics v20, EViews v10, and MATLAB R2019a.

## 4. Results

### 4.1. Descriptive Statistics and Maps

Table 1 presents the main descriptive indicators for the four analyzed variables, specifically the minimum value, maximum value, mean, and std. deviation.

**Table 1.** Descriptive statistics.

|  | N | Minimum | Maximum | Mean | Std. Deviation |
|---|---|---|---|---|---|
| Big_Data | 107 | 6 | 38 | 18.80 | 7.138 |
| E_Commerce_Sales | 105 | 9 | 60 | 32.24 | 11.644 |
| CRM_Mk | 107 | 5 | 30 | 14.01 | 5.694 |
| CRM_Elsewhere | 108 | 3 | 38 | 16.97 | 7.054 |
| Valid N (Listwise) | 104 | | | | |

In order to enhance a better observation of the data that was analyzed, we are proposing two sets of maps, one corresponding to 2014 and one to 2021, exposing the national percentages of big data, e-commerce sales, and CRM software usage for the EU 28 area.

From Figures 2 and 3, one can identify a slight increase in all of the analyzed variables, as for the averages and for the maximum values, while countries with the highest growth in all three variables were Sweden, Estonia, Czech Republic, the Netherlands, Belgium, and Greece.

While Poland, France, and Italy presented significant increases in big data and e-commerce sales usage, Romania registered a significant increase in e-commerce sales.

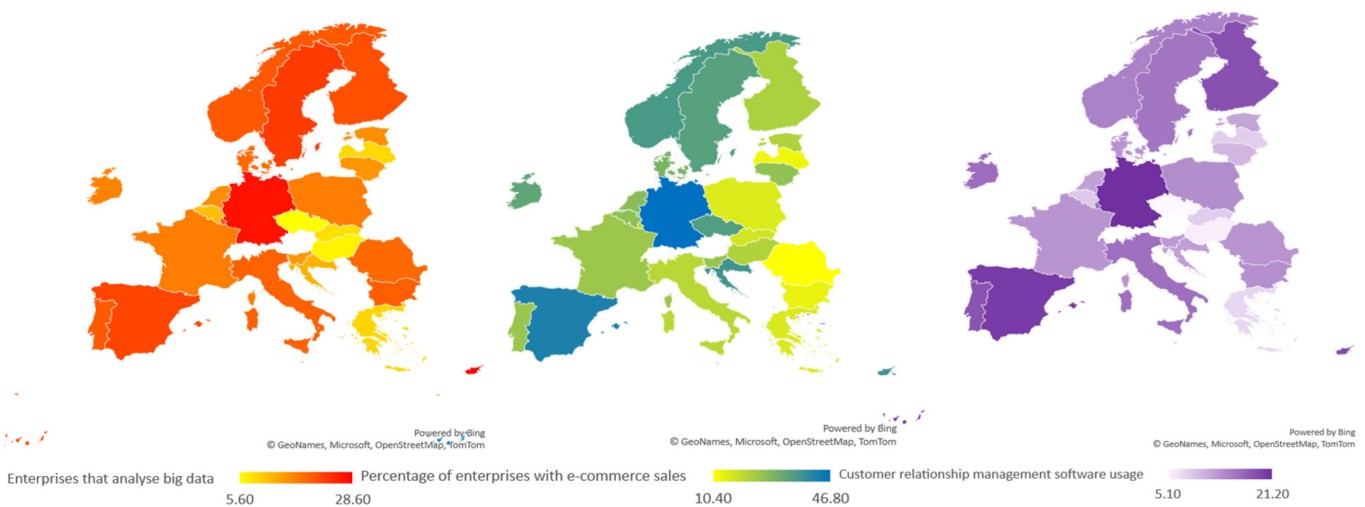

**Figure 2.** Enterprises that analyze big data, percentage of enterprises with e-commerce sales, customer relationship management software for marketing in 2014.

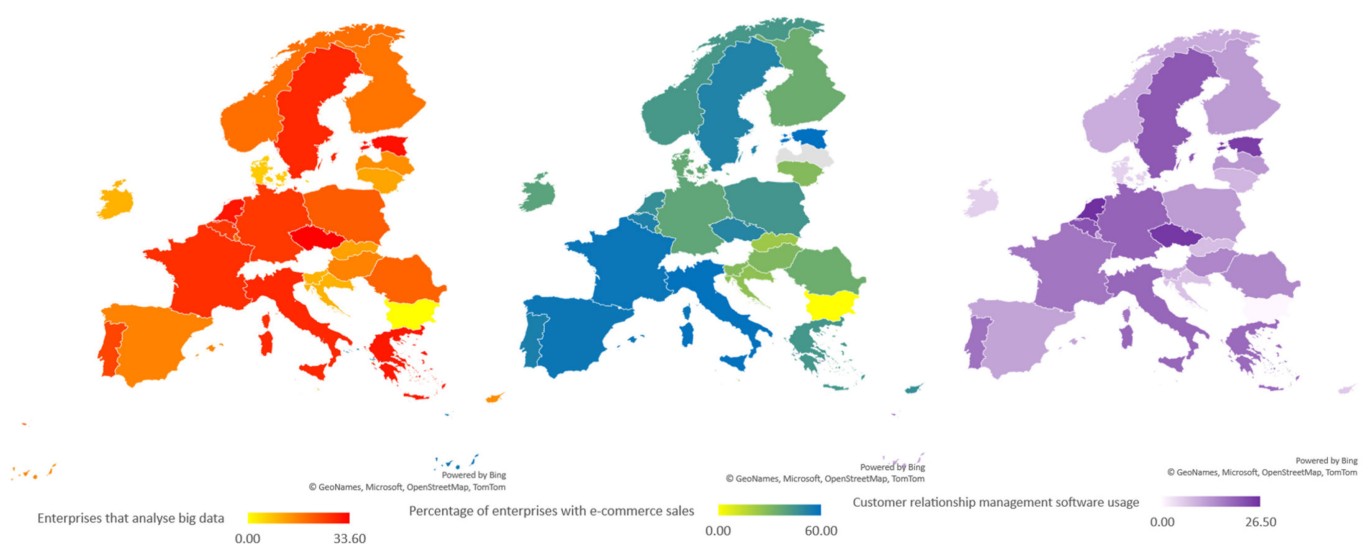

**Figure 3.** Enterprises that analyze big data, percentage of enterprises with e-commerce sales, customer relationship management software for marketing in 2021.

*4.2. Testing for Autocorrelation and Stationarity*

To test the autocorrelation, the Durbin–Watson procedure was performed. For the Durbin–Watson procedure, the null and alternative hypothesis were

- H0: There is no correlation among the residuals.
- HA: The residuals are autocorrelated.

Although our test statistic of 1.763 does lie outside of this range, as seen in Table 2, we may consider that test statistic values in the range of 1.5 to 2.5 are relatively normal, while the one that falls outside of this range could be a definite cause for concern. Thus, we can conclude that there is no correlation among the residuals.

**Table 2.** Results of the Durbin–Watson procedure.

| Model | Durbin–Watson |
|---|---|
| T-stat | 1.763 |
| Lower critical value | 1.482 |
| Upper critical value | 1.604 |
| k | 3 |
| N | 100 |

a. Predictors: (Constant), CRM_ELSEWHERE, CRM_MK, BIG_DATA, b. Dependent Variable: E_COMMERCE_SALES.

In Table 3, results computed under the Dickey–Fuller test (ADF) procedure are presented. Tested hypotheses are

H0: A unit root is present in a time series sample.
H1: The time-series sample has no unit root; thus, it is stationary.

**Table 3.** Results of the ADF procedure.

| Model | ADF—Fisher Chi-Square | *p*-Value | ADF—Choi Z-Stat | *p*-Value |
|---|---|---|---|---|
| Variable | Statistic | | Statistic | |
| Big data usage | 49.4326 | 0.0000 | −5.68702 | 0.0000 |
| CRM usage for marketing | 48.2818 | 0.0000 | −5.57162 | 0.0000 |
| CRM usage for other activities | 57.2026 | 0.0000 | −6.31778 | 0.0000 |
| E-commerce sales usage | 25.5744 | 0.0012 | −3.43753 | 0.0003 |

In Table 3, we can observe that all of the *p*-values are less than the significance level of 0.5; thus, we can reject the null hypothesis and conclude that the time series are stationary.

Next, we have constructed a 3-dimensional plot to enhance the observation of the relationships that were formed among the analyzed variables, as it can be seen in Figure 4. On the X-axis, the e-commerce usage is presented; on the Y-axis, big data usage is presented; while on the Z-axis, the CRM software usage for marketing activities is constructed. Thus, it can be observed that a high usage of big data is positively associated with a high usage of e-commerce sales. Also, CRM usage is positively correlated with e-commerce sales usage.

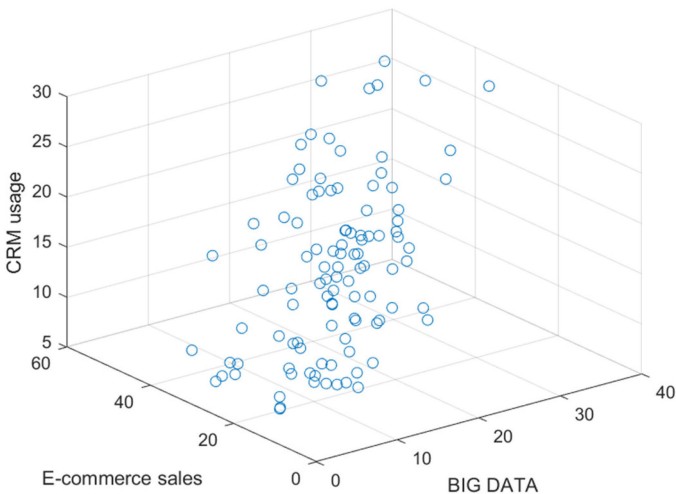

**Figure 4.** Three-dimensional plot of big data, CRM and e-commerce sales.

### 4.3. F-Test for the Equality of Variances

In order to test if the variances between the groups prepared for the hypothesis testing are equal, we have performed some F-tests for equality of the variances. Results of the F-test are shown in Tables 4–6.

**Table 4.** F-test two-sample for variance, the impact of big data usage on the percentage of companies using e-commerce sales.

|  | Variable 1 | Variable 2 |
|---|---|---|
| Mean | 27.3549 | 37.04906 |
| Variance | 124.8877 | 103.5233 |
| Observations | 51 | 53 |
| df | 50 | 52 |
| F | 1.206373 |  |
| P(F ≤ f) one-tail | 0.251976 |  |
| F Critical one-tail | 1.590168 |  |

**Table 5.** F-test two-sample for variances, the impact of customer relationship management software for marketing on the percentage of e-commerce sales.

|  | Variable 1 | Variable 2 |
|---|---|---|
| Mean | 27.56275 | 36.84906 |
| Variance | 116.5236 | 115.4349 |
| Observations | 51 | 53 |
| df | 50 | 52 |
| F | 1.009432 |  |
| P(F ≤ f) one-tail | 0.485966 |  |
| F Critical one-tail | 1.590168 |  |

**Table 6.** F-test two-sample for variances, the impact of customer relationship management software for elsewhere on the percentage of e-commerce sales.

|  | **Variable 1** | **Variable 2** |
|---|---|---|
| Mean | 27.67451 | 36.64528 |
| Variance | 112.0707 | 123.1883 |
| Observations | 51 | 53 |
| df | 50 | 52 |
| F | 0.909752 |  |
| P(F ≤ f) one-tail | 0.369185 |  |
| F Critical one-tail | 0.627324 |  |

Hypothesis tested are

- H0: The two variances are equal.
- H1: The two variances are not equal.

In Table 4, the *p*-value (0.251) is higher than the standard significance level of 0.05, and we cannot reject the null hypothesis. Our sample data support the hypothesis that the population variances are equal.

In Table 5, the *p*-value (0.251) is higher than the standard significance level of 0.05, and we cannot reject the null hypothesis. Our sample data support the hypothesis that the population variances are equal.

In Table 6, the *p*-value (0.251) is higher than the standard significance level of 0.05, and we cannot reject the null hypothesis. Our sample data support the hypothesis that the population variances are equal.

*4.4. t-Test Assuming Equal Variances*

Next we are using using the Student's T-test to determine if two population means are equal.

The hypotheses tested are presented as follows.

Case 1:

- H0: The usage of big data has no significant influence on the percentage of e-commerce sales.
- H1: The usage of big data has a significant influence on the percentage of e-commerce sales.

Case 2:

- H0: The usage of CRM for marketing has no significant influence on the percentage of e-commerce sales.
- H1: The usage of CRM for marketing has a significant influence on the percentage of e-commerce sales.

Case 3:

- H0: The usage of CRM elsewhere has no significant influence on the percentage of e-commerce sales.
- H1: The usage of CRM elsewhere has a significant influence on the percentage of e-commerce sales.

The results of the t-test assuming equal variances are presented in Tables 7–9.

**Table 7.** Testing the impact of big data on the percentage of e-commerce sales.

|  | **Variable 1** | **Variable 2** |
|---|---|---|
| Mean | 27.35490196 | 36.85741 |
| Variance | 124.8877255 | 103.5534 |
| Observations | 51 | 54 |
| Pooled Variance | 113.9098865 |  |
| Hypothesized Mean Difference | 0 |  |
| df | 103 |  |

**Table 7.** *Cont.*

|  | Variable 1 | Variable 2 |
| --- | --- | --- |
| t Stat. | −4.559788889 | |
| P(T ≤ t) one-tail | 0.00 | |
| t Critical one-tail | 1.66 | |
| P(T ≤ t) two-tail | 0.00 | |
| t Critical two-tail | 1.98 | |

**Table 8.** Testing the impact of customer relationship management software for marketing on the percentage of e-commerce sales.

|  | Variable 1 | Variable 2 |
| --- | --- | --- |
| Mean | 27.5627 | 36.6611 |
| Variance | 116.5236 | 115.1643 |
| Observations | 51.0000 | 54.0000 |
| Pooled Variance | 115.8242 | |
| Hypothesized Mean Difference | 0.0000 | |
| df | 103.0000 | |
| t Stat. | −4.3296 | |
| P(T ≤ t) one-tail | 0.0000 | |
| t Critical one-tail | 1.6598 | |
| P(T ≤ t) two-tail | 0.0000 | |
| t Critical two-tail | 1.9833 | |

**Table 9.** Testing the impact of customer relationship management software elsewhere on the percentage of e-commerce sales.

|  | Variable 1 | Variable 2 |
| --- | --- | --- |
| Mean | 27.6745098 | 36.555556 |
| Variance | 112.0707373 | 121.29874 |
| Observations | 51 | 54 |
| Pooled Variance | 116.8191281 | |
| Hypothesized Mean Difference | 0 | |
| df | 103 | |
| t Stat. | −4.208181333 | |
| P(T ≤ t) one-tail | 0.0000 | |
| t Critical one-tail | 1.6598 | |
| P(T ≤ t) two-tail | 0.0001 | |
| t Critical two-tail | 1.9833 | |

Normally, we say that a *p*-value of 0.05 or less is significant, in which case we reject the null hypothesis (accept the alternative hypothesis). In our case, presented in Table 7, two groups were compared, one counting the e-commerce usage in countries that have less than 20% usage in big data and another group that counts e-commerce usage for countries in which more than 20% of the enterprises use big data. Because the *p*-value is less than alpha, we can reject the null hypothesis, thus concluding that the usage of big data has a significant influence on the percentage of e-commerce sales.

In the case presented in Table 8, two groups were compared, one counting the e-commerce usage in countries that have less than 20% usage in CRM software for marketing activities and another group that counts e-commerce usage for countries in which more than 20% of the enterprises use CRM software for marketing activities. Because the *p*-value is less than alpha, we can reject the null hypothesis, thus concluding that the usage of CRM for marketing has a significant influence on the percentage of e-commerce sales.

In the case presented in Table 9, two groups were compared, one counting the e-commerce usage in countries that have less than 20% usage in CRM software for other

activities, except marketing, and another group that counts e-commerce usage for countries in which more than 20% of the enterprises use CRM software for other activities, except marketing. Because the *p*-value is less than alpha, we can reject the null hypothesis, thus concluding that the usage of CRM elsewhere has a significant influence on the percentage of e-commerce sales.

### 4.5. Correlation Matrix

In order to perform a multiple regression, we first analyzed the Pearson correlations among e-commerce sales, big data usage, and customer relationship usage in marketing or in other activities. It can be observed from Table 10 that all three analyzed variables are significantly correlated, with a significance level under 0.01. Also, all of the correlations computed are positive.

**Table 10.** Pearson correlation matrix between big data usage, e-commerce sales usage, customer relationship management software usage in marketing activities or in other activities, except marketing.

| | Correlations | | | | |
|---|---|---|---|---|---|
| | | **Big_Data** | **e_Commerce** | **CRM_Marketing** | **CRM_Elsewhere** |
| big_data | Pearson Correlation | 1 | 0.514 ** | 0.921 ** | 0.966 ** |
| | Sig. (2-tailed) | | 0.000 | 0.000 | 0.000 |
| | N | | 104 | 107 | 107 |
| e_commerce | Pearson Correlation | | 1 | 0.460 ** | 0.457 ** |
| | Sig. (2-tailed) | | | 0.000 | 0.000 |
| | N | | | 104 | 105 |
| CRM_maarketing | Pearson Correlation | | | 1 | 0.866 ** |
| | Sig. (2-tailed) | | | | 0.000 |
| | N | | | | 107 |
| CRM_elsewhere | Pearson Correlation | | | | 1 |
| | Sig. (2-tailed) | | | | |
| | N | | | | |

** Correlation is significant at the 0.01 level (2-tailed).

### 4.6. Multivariate Regression

In order to predict the usage of e-commerce sales based on the usage of big data and CRM, we have performed a multivariate linear regression, constructed as presented in equation 1. Coefficients were estimated using the least-squares method. The results, presented in Table 11, show significant coefficients for big data usage and usage of CRM for other activities than marketing, while insignificant coefficients for CRM software usage for marketing activities. Also, at a one percent modification in big data usage, e-commerce usage will be modified by 2.3 percentage points in the same direction. In contrast, for a one percentage point modification for CRM software used for activities other than marketing, e-commerce usage modifies by 1.2 percentage points in the contrary direction.

**Table 11.** Estimation of the coefficients for the multivariate linear regression.

| Model | | Unstandardized Coefficients | | Standardized Coefficients | t | Sig. |
|---|---|---|---|---|---|---|
| | | **B** | **Std. Error** | **Beta** | | |
| 1 | (Constant) | 15.390 | 2.800 | | 5.497 | 0.000 |
| | BIG_DATA | 2.324 | 0.710 | 1.431 | 3.273 | 0.001 |
| | CRM_MK | −0.426 | 0.458 | −0.208 | −0.930 | 0.354 |
| | CRM_ELSEWHERE | −1.223 | 0.560 | −0.749 | −2.182 | 0.031 |

a. Dependent Variable: E_COMMERCE_SALES.

In Tables 12 and 13, the goodness of fit and ANOVA analysis for the multivariate linear regression are presented. An R-squared of 0.299 is observed, while an adjusted R-squared of 0.278 is observed.

**Table 12.** Goodness of fit.

| Model | R | R Square | Adjusted R Square | Std. Error of the Estimate |
|---|---|---|---|---|
| 1 | 0.547 | 0.299 | 0.278 | 9.932 |

a. Predictors: (Constant), CRM_ELSEWHERE, CRM_MK, BIG_DATA, b. Dependent Variable: E_COMMERCE_SALES.

**Table 13.** ANOVA.

| | Model | Sum of Squares | df | Mean Square | F | Sig. |
|---|---|---|---|---|---|---|
| | Regression | 4205.098 | 3 | 1401.699 | 14.209 | 0.000 |
| 1 | Residual | 9864.990 | 100 | 98.650 | | |
| | Total | 14070.088 | 103 | | | |

a. Dependent Variable: E_COMMERCE_SALES, b. Predictors: (Constant), CRM_ELSEWHERE, CRM_MK, BIG_DATA.

The multivariate linear regression analysis embarked on a detailed journey to understand the intricate relationships between e-commerce sales and its potential drivers, namely big data usage and CRM applications in marketing and other domains. The depth of knowledge obtained from this analysis holds considerable significance for e-commerce strategies and how technology is integrated.

Regarding the influence of big data, a noteworthy revelation emerges. As big data usage sees a rise, so does e-commerce sales, with a 2.3 percentage point increase in the latter for every one percentage point growth in the former. This observation not only highlights the immense power of big data in propelling digital sales but also resonates with the prevalent sentiment on the importance of data-driven decisions in the modern business scenario.

Venturing into the realm of CRM, we observe a contrasting scenario. On the one hand, when CRM is utilized for functions beyond marketing, termed here as CRM_ELSEWHERE, there appears to be a diminishing effect on e-commerce sales. Specifically, for every percentage point increase in CRM_ELSEWHERE usage, e-commerce sales decrease by 1.2 percentage points. On the contrary, the impact of using CRM specifically for marketing, termed CRM_MK, does not showcase any significant influence on e-commerce sales. This divergence raises a crucial point: while CRM tools are undeniably beneficial in certain areas, their blanket application might not guarantee positive outcomes, especially in the realm of e-commerce sales.

Delving deeper into the analysis, the model's goodness-of-fit metrics, such as the R and adjusted R-squared values, shine a light on its explanatory prowess. They suggest that the model can account for approximately 27.8% to 29.9% of e-commerce sales fluctuations. However, this also points to the fact that significant variability in e-commerce sales can be attributed to variables not present in this model. Moreover, the ANOVA results lend further credibility to the model by confirming the value of at least one predictor in forecasting e-commerce sales, further emphasized by a meaningful F-statistic.

Taking a step back to interpret these findings, it becomes evident that while big data and CRM are compelling tools, their deployment needs to be both strategic and tailored to specific scenarios. The negative coefficient associated with CRM_ELSEWHERE is particularly thought-provoking, prompting businesses to question if an over-dependence on technology might be overshadowing crucial human touchpoints in sales or perhaps diverting critical resources from primary sales-driven activities. Furthermore, it is imperative for businesses to acknowledge the vast array of external factors, such as market trends, competitive elements, evolving consumer behaviors, and wider economic indicators, that possibly influence e-commerce sales. Adopting a comprehensive approach in strategy development becomes vital.

Drawing this exploration to a close, it is evident that the relationship between technology use and e-commerce outcomes is a delicate dance. Tools like big data and CRM can indeed be game-changers. However, understanding their nuanced impacts within the e-commerce framework demands a more profound and context-sensitive examination.

## 5. Discussion

The primary objective of our research was to discern the potential influence of contemporary technological tools, such as big data and CRM, on e-commerce sales across different countries. Our findings not only confirm the significant role of big data in bolstering e-commerce outcomes but also shed light on the more intricate relationship between CRM tools and e-commerce sales.

Starting with the univariate analysis, it became clear that countries utilizing big data and CRM tools, both for marketing and other activities, showcased a distinct advantage in terms of e-commerce sales. The statistical significance in the results gives credence to the popular belief that technological advancement, particularly in the realm of data and customer relationship management, plays a pivotal role in bolstering e-commerce outcomes.

However, while the univariate analysis provides a broad-brush understanding, the multivariate regression analysis allows us to delve deeper into these relationships, teasing out more nuanced insights.

Our regression analysis was especially revelatory in terms of the positive and significant relationship between big data usage and e-commerce sales. The observed phenomenon that a marginal increase in big data usage leads to a proportionally larger rise in e-commerce sales underscores the strategic value of data-driven decision-making in the contemporary business landscape. This positive correlation suggests that as enterprises increasingly rely on analytics and data-driven insights, they can potentially harness greater e-commerce sales, thereby amplifying their digital footprint.

On the other hand, the relationship between CRM tools and e-commerce sales manifested in an intriguing dichotomy. While the usage of CRM for functions other than marketing seemed to negatively influence e-commerce sales, its usage specifically for marketing did not showcase a statistically significant impact on e-commerce outcomes. This draws attention to the importance of strategic technology deployment. It is imperative for businesses to understand that the indiscriminate application of tools like CRM may not always yield the desired results, particularly concerning e-commerce sales. The negative relationship of CRM usage for non-marketing functions with e-commerce sales warrants further investigation. Potential hypotheses could be that an over-reliance on technology might be diluting essential human touchpoints or that resources allocated to these functions could be better utilized elsewhere.

Furthermore, the model's explanatory statistics indicate that while big data and CRM applications hold considerable sway, a significant portion of the e-commerce sales variability is shaped by other external factors. This could encompass aspects like market dynamics, competitive pressures, and changing consumer behaviors. Thus, when strategizing for e-commerce, businesses need to adopt a more holistic approach, taking into account the broader ecosystem in which they operate.

While the positive impact of big data on e-commerce sales might align with prevalent sentiments, our findings regarding CRM's dichotomous impact could surprise a knowledgeable reader. Previous research has often championed the benefits of CRM tools in enhancing customer relationships and driving sales. Our research nuances this narrative, suggesting that while CRM tools are beneficial, their indiscriminate application might not guarantee positive outcomes in e-commerce sales.

Our findings, which emphasize the undeniable significance of big data in shaping e-commerce outcomes and the nuanced relationship between CRM and e-commerce sales, contribute to the existing body of knowledge in several ways.

Firstly, while previous studies have highlighted the role of big data in business strategies, our research offers a more granular understanding of its direct impact on e-commerce sales. For instance, a study by [4] revealed that CRM with big data has enabled businesses to become more aggressive in terms of marketing strategies, such as push notifications to their potential target audiences. Our study builds on this by quantifying the relationship, suggesting that a marginal increase in big data usage leads to a proportionally larger rise in e-commerce sales.

Furthermore, our research unveils a dichotomous impact of CRM on e-commerce sales. While [6] proposed a theoretical model showing the combined impact of big data and CRM capabilities on an organization's strategic sales performance, our findings provide empirical evidence of the contrasting effects of CRM when used for marketing versus other functions. This distinction is crucial for businesses aiming to optimize their CRM strategies for e-commerce growth.

Moreover, the nuanced relationship we identified between CRM's application in non-marketing functions and its potential negative impact on e-commerce sales resonates with the findings of [12]. Their study cautioned e-vendors about the potential negative applications of over-relying on big data analytics. Our research further elucidates this by highlighting the specific areas where CRM might not be as beneficial, emphasizing the need for businesses to be discerning in their deployment of technology.

In essence, our research not only corroborates existing findings but also introduces novel insights into the dynamic interplay between technology and e-commerce. By offering a more detailed and quantified perspective on the relationships between big data, CRM, and e-commerce sales, we believe our work distinguishes itself from many other similar studies, providing both academic and practical value.

In the realm of e-commerce, our research offers several practical implications that can guide businesses in optimizing their strategies. By dissecting the intricate relationships between e-commerce sales and its potential drivers, namely big data usage and CRM applications, we provide actionable insights for businesses operating in the digital landscape.

Our findings are particularly relevant for e-commerce businesses, digital marketing professionals, and decision-makers in organizations that are at the crossroads of technology adoption. For instance, the positive correlation we identified between big data usage and e-commerce sales underscores the strategic value of data-driven decision-making. Businesses can leverage this insight by investing more in big data analytics tools and training, ensuring that their strategies are informed by real-time data and insights.

Consider an e-commerce platform that is contemplating whether to invest in big data analytics. Our research suggests that such an investment could lead to a proportionally larger rise in sales. By analyzing customer behavior, preferences, and purchase patterns, the platform can offer personalized product recommendations, leading to increased sales and customer retention.

On the flip side, our research highlights the nuanced impact of CRM tools. While CRM's role in marketing did not show a significant impact on e-commerce sales, its application elsewhere seemed to have a counterintuitive effect. This insight is invaluable for businesses that might be considering a blanket application of CRM tools. Instead of indiscriminately deploying CRM across all functions, businesses should be more strategic, focusing on areas where CRM can genuinely add value.

The intricate relationship between big data usage and e-commerce sales has been a focal point of numerous studies in recent years. Our research, which highlighted the positive correlation between big data usage and e-commerce sales, is supported by a study conducted on 122 e-business companies in Maharashtra, India. This study concluded that big data are instrumental in analyzing customer behavior, systematic analytics, and gaining a competitive advantage [36].

Further, a cross-cultural study by [37] revealed that the impact of big data usage varies across countries. For instance, in the USA, big data-embedded product testing had the highest effect on sales growth and gross margin, while in Australia and the UK, big data-embedded commercialization showed the most significant impact on sales growth and gross margins. This suggests that the influence of big data on e-commerce sales is not only significant but also varies based on regional factors and the specific application of big data.

Another study by [7] emphasized the positive moderating effect of big data analytics capability. They found that the stronger this capability, the greater the impact of big data

on sales growth and gross margin. This underscores the importance of not just using big data but harnessing it effectively to drive sales.

Interestingly, while our research found a dichotomous impact of CRM on e-commerce sales, a study by [6] proposed a unique theoretical model that showed the combined impact of big data and CRM capabilities on an organization's strategic sales performance. This suggests that the interplay between big data and CRM can be complex and warrants further exploration.

Moreover, the negative influence of CRM in non-marketing functions, as highlighted in our research, resonates with the findings of a study by [12], which cautioned e-vendors about the potential negative applications of over-relying on big data analytics. This reiterates the need for businesses to be discerning in their deployment of technology.

Thus, the myriad of studies in this domain underscores the profound implications of big data and CRM on e-commerce sales. While the positive impact of big data is undeniable, the nuanced relationship between CRM and e-commerce sales necessitates a deeper, more contextual exploration. As businesses continue to navigate the digital landscape, strategic and judicious deployment of these technologies will be paramount.

Thus, in an era where every investment decision is important, our research may encourage businesses to make informed choices, maximizing their returns and staying ahead in the competitive e-commerce domain.

## 6. Limitations and Potential Future Research Directions

Our study, while providing valuable insights into the relationship between the usage of big data, CRM, and e-commerce sales, comes with its set of constraints. One primary limitation was the reliance on data from Eurostat. Although Eurostat is a robust source, its data might not encapsulate the nuanced differences that exist in individual country contexts or the diverse verticals within e-commerce. This limitation is compounded by potential biases or gaps in the data collection process, which might influence the generalizability of our findings.

Additionally, the temporal scope of our research needs to be considered. The rapid pace of technological changes and shifts in e-commerce trends means that our study serves as a snapshot in time. As these dynamics evolve, longitudinal studies spanning longer durations might offer deeper insights into the changing nature of these relationships.

A significant aspect to ponder upon is the simplification of variables in our research. By relying on distinctions such as 'less than 20% usage' versus 'more than 20% usage', we might inadvertently be oversimplifying the spectrum of technology adoption. This approach might obscure the intricate relationships that exist on a gradient rather than within these somewhat arbitrary dichotomous brackets.

Furthermore, our study's focus on big data and CRM tools, while significant, does not encompass the myriad factors influencing e-commerce sales. From macroeconomic indicators to cultural nuances, various elements play a role in e-commerce dynamics, and not accounting for these variables might have confounded the relationships we observed.

The regional concentration of our dataset is another facet that merits attention. Our primary reliance on Eurostat data may have limited the global applicability of our findings. Different regions or countries could have unique dynamics that our study did not capture.

Looking ahead, the academic and business community has a vast expanse of potential research opportunities. Incorporating data from sources beyond Eurostat could paint a more comprehensive global picture, helping us discern regional nuances. A more granular examination of individual e-commerce sectors could reveal sector-specific insights that our broader approach might have missed. Moreover, with emerging technologies like AI and blockchain reshaping the digital commerce landscape, it would be enlightening to assess their influence on e-commerce. A blend of qualitative research could complement our quantitative findings, elucidating the 'why' behind observed trends. Lastly, observing these dynamics over more extended periods will reveal long-term impacts and the evolving nature of technological interventions in e-commerce.

In summary, while our research has paved the way for a more profound understanding of the interplay between technology and e-commerce, there remains a vast landscape to explore, refine, and understand in this domain.

## 7. Conclusions

Thus, data science solutions offer businesses a competitive edge by providing them with actionable insights derived from data. These insights, when effectively leveraged, can lead to improved operational efficiency, enhanced customer experiences, and, ultimately, increased revenues. As businesses continue to operate in an increasingly data-driven environment, the importance of data science solutions in driving revenue growth cannot be overstated.

Our research ventured into this domain, aiming to illuminate the influence of contemporary technological tools, notably big data and CRM, on e-commerce sales across an array of countries. By leveraging both univariate analyses and a deeper dive through multivariate regression, we strived to shed light on the nuances of these relationships and extract meaningful insights that could guide strategic decision-making in the realm of e-commerce.

Our findings emphasize the undeniable significance of big data in shaping e-commerce outcomes. With a clear and positive relationship established, it is evident that data-driven strategies can act as catalysts for digital sales growth. This stands as a testament to the increasing importance of analytical competencies in today's business world, where data are often heralded as the 'new oil'.

Contrastingly, the results concerning CRM tools highlighted a more intricate narrative. While CRM's role in marketing did not show a significant impact on e-commerce sales, its application elsewhere seemed to work counterintuitively, pointing to the necessity of strategic and judicious use of such tools. This bifurcation suggests that it is not just the adoption but the application of technology that determines its efficacy in boosting e-commerce outcomes.

At its core, our research underscores the symbiotic relationship between technology and e-commerce, albeit with certain caveats. While tools like big data and CRM have the power to reshape the e-commerce trajectory, their impact is contingent on the broader business ecosystem and the strategic nuances of their deployment. As digital platforms continue to burgeon, businesses need to cultivate a discerning approach to technology adoption, ensuring alignment with overarching objectives and market dynamics.

Moreover, our findings underscore the profound implications of big data in the e-commerce domain. As businesses increasingly pivot towards data-driven strategies, our research reaffirms the strategic value of such an approach. The positive correlation between big data usage and e-commerce sales emphasizes the transformative power of data analytics in today's business world. This not only aligns with the prevailing sentiment about the importance of data but also adds depth to our understanding of how data-driven insights can be harnessed to drive e-commerce growth.

Conversely, our exploration into the realm of CRM revealed a more complex narrative. While the benefits of CRM tools, especially in enhancing customer relationships, have been widely championed, our research nuances this understanding. We highlighted the dichotomous impact of CRM on e-commerce sales, suggesting that while these tools hold immense potential, their indiscriminate application might not always yield the desired outcomes. This insight is particularly crucial for businesses navigating the digital landscape, ensuring that their technological investments align with their overarching objectives.

In terms of theoretical novelty, our research distinguishes itself by delving deeper into the relationships between e-commerce sales and its potential drivers. While previous studies have often approached these relationships in isolation, our comprehensive approach allowed us to tease out more intricate insights. The contrasting findings regarding big data and CRM's impact on e-commerce sales offer a fresh perspective, challenging prevailing notions and enriching the academic discourse in this domain.

However, our reliance on Eurostat data might not capture the nuances of individual countries or diverse e-commerce sectors, potentially introducing biases. The temporal scope of our study, representing a specific moment in the rapidly evolving digital landscape, calls for more extended, longitudinal research. Our categorization approach, distinguishing based on usage percentages, might have oversimplified technology adoption nuances. While we focused on big data and CRM tools, myriad other factors influencing e-commerce were not considered, and our Eurostat-centric dataset might limit the global applicability of our findings. Despite these constraints, our study has opened doors for deeper exploration in the realm of technology and e-commerce, highlighting the need for diversified data sources, sector-specific studies, and the inclusion of emerging technologies.

As we move further into the digital age, our results highlight the complex relationships between data science solutions in the e-commerce space in the EU's accommodation and food service sector, which may help them understand the many drivers to focus on in order to increase company turnover.

**Author Contributions:** Conceptualization, M.P.C., D.A.M., L.C.C., R.A.N., A.B. and S.-V.O.; Methodology, M.P.C., D.A.M., L.C.C., R.A.N., A.B. and S.-V.O.; Supervision, M.P.C.; Writing—original draft, M.P.C., D.A.M., L.C.C., R.A.N., A.B. and S.-V.O.; Writing—review and editing, M.P.C., D.A.M., L.C.C., R.A.N., A.B. and S.-V.O. All authors have read and agreed to the published version of the manuscript.

**Funding:** This research received no external funding.

**Data Availability Statement:** Used data are available at the following url: https://ec.europa.eu/eurostat/databrowser/view/ISOC_EC_ESELN2__custom_7176858/default/table, https://ec.europa.eu/eurostat/databrowser/view/ISOC_EB_BDN2__custom_7625275/default/table?lang=en, https://ec.europa.eu/eurostat/databrowser/view/ISOC_EB_BDN2__custom_7625378/default/table?lang=en, https://ec.europa.eu/eurostat/databrowser/view/ISOC_EB_IIPN2__custom_7625373/default/table?lang=en, https://ec.europa.eu/eurostat/databrowser/view/ISOC_EB_IIPN2__custom_7625375/default/table?lang=en, accessed on 7 August 2023.

**Conflicts of Interest:** The authors declare no conflict of interest.

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
