# Peer review of "The Impact of Data Science Solutions on the Company Turnover"

_information, doi:10.3390/info14100573_

Round 1
Reviewer 1 Report
I am pleased to have the opportunity to review this research paper.
This study analyzes Eurostat data from 2014 to 2021 and about e-commerce sales in different countries. It shows the potential role of CRM and data science solutions in improving business performance in the EU's Accommodation and Food Service industry.
Тhe main contribution of the authors is in the proposed analysis methodology.
The topic of the Article is suitable for the magazine. The logic of the presentation is good, the recommended structure for articles is followed.
The cited literary sources are mostly relevant and up-to-date.
I highly appreciate the carried out research, the approbation and the analysis of the obtained results.
In order to improve the article, I recommend that the authors more clearly formulate the purpose of the research in the introduction, indicate what their contributions are and how their research is useful.
Author Response
First of all, thank you for your revision!
In the revised introduction, we have explicitly stated our primary research question: "To what extent do contemporary technological tools, specifically big data and CRM, influence e-commerce sales across different countries?"
We have also elaborated on the contributions and utility of our research. Our study underscores the nuanced relationship between technology and e-commerce. We found that while big data and CRM can significantly influence e-commerce, their impact is intricately tied to their strategic application and the broader business ecosystem. Our findings offer businesses insights into cultivating a discerning approach to technology adoption, ensuring that it aligns with overarching objectives and market dynamics.
In terms of novelty, our research distinguishes itself by offering intricate insights into the relationships between e-commerce sales and its potential drivers, enriched by a comprehensive approach that examines these relationships in unison rather than isolation.
Please see the attachment. We also highlighted the added text.

Reviewer 2 Report
Dear Authors,
thank you for the article. The authors performed an analysis of the problem area and summarized the obtained results. The article lacks a more detailed description of the data set used, performed evaluating and obtained results. Providing more details in mentioned areas would be beneficial for the readers and increase the overall level of the article.
The manuscript contains few typos but these do not affect the overall quality of the article.
There are some comments that I hope, can be beneficial to the article:
- 1/24 (page 1/line 24) – it would be appropriate to provide a link where it is possible to obtain the input data set
- 3/101, 102, 108 … – please, describe all used acronyms when they first occur, for example ECCRM, B2C, e-CRM…
- 5/207 – it would be appropriate to specify more precisely the input data set
- 5/208 – it would be appropriate to state more precisely on what basis the dependent and independent variables were selected
- 5/203 – the description of the individual parameters of the equation is missing
- 6/246 – Figure 2 is very small and difficult to read, it would be appropriate to enlarge it (or display each part of figures separately) to make it easier to read
- 6/256 – similarly also Figure 3 is very small and difficult to read
- 7/281 – you state correlations between parameters but do not present their correlation matrix, it would be appropriate to include it here
- 9/332, 10/340, 10/348 – probability parameter is usually denoted “alpha” instead of “alfa”
Author Response
Thank you for your detailed feedback and constructive comments on our manuscript. We are pleased to inform you that we have addressed each of the points you raised to enhance the quality and clarity of our paper.
We have included the links to the datasets in the Data Availability Statement section.
All acronyms, including ECCRM, B2C, e-CRM, etc., have been described when they first occur in the text to enhance readability and comprehension.
We have provided a more precise specification of the input data set and elaborated on the basis upon which the dependent and independent variables were selected. The datasets and variables have been described in detail to offer readers a comprehensive understanding of the data and the rationale behind their selection.
The description of the individual parameters of the equation has been added to ensure that readers have a clear understanding of each parameter’s role and significance in the analysis.
We have also enlarged Figures 2 and 3 to enhance their readability and ensure that readers can easily interpret the data presented.
The correlation matrix has been included to provide a detailed view of the correlations between parameters. In order to perform a multiple regression, we firstly analyzed the Pearson correla-tions among e-commerce sales, big data usage and customer relationship usage in marketing or in other activities. It can be observed, in table 10, that all the three an-alyzed variables are significantly correlated, with a significance level under 0.01. Also, all of the correlations computed are positive.
Lastly, we have corrected the term "alfa" to "alpha" to adhere to the conventional notation and enhance the paper’s professionalism and readability.
Please see the attachment. We also highlighted the added text.

Reviewer 3 Report
.
In the introduction, what key theoretical perspectives and empirical findings in the main literature have already informed the problem formulation? What major, unaddressed puzzle, controversy, or paradox does this research address?
Why does it need to be addressed?
Why it should be now - not in the past?
Further, in the introduction, what is the recent knowledge gap of the main literature that the author needs to write this research? What we have known and what we have not known? What is missing from current works? Please explain and give examples!
In terms of the knowledge gap, it will be best if the research challenge/knowledge gap could be stated in one article or more articles in the main literature (optional). Assure that you have included all key articles (e.g., most widely cited articles) in the main literature. Mention them.
Theoretical contribution to the main literature seems to be weak. How does the research fulfil the knowledge gap discussed in the introduction? In terms of the knowledge gap, indicates what is the new, exciting, and not trivial contribution you offer.
In terms of theoretical contribution, show the theoretical novelty that your work offers. How does this novelty distinguish your work from other similar works?
Practical Implications seem to be unclear? Please mention and make a reference! Further, please answer for whom it is relevant?
As to practical implications, how do the findings help these people to solve their daily problems?
Please explain and give examples! Assure that any recommendation is clear and actionable for organisations.
Research question must be explicitly stated in the introduction.
Show how your findings are addressing your research question.
Show how your finding could surprise a knowledgeable reader in the main literature. For example, what new things that your findings offer? Is there anything from previous research that your finding refutes? If there is nothing that is rejecting previous research’s findings, why do you write this paper?
As to practical implications, show how your recommendation is timely.
What are the limitations of the present work?
Conclusion is too short. Add more explanation.
.
Author Response
Thank you for your detailed feedback and constructive comments on our manuscript. We are pleased to inform you that we have addressed each of the points you raised.
Introduction and Theoretical Perspectives:
We have enriched the introduction by clearly stating our primary research question: "To what extent do contemporary technological tools, specifically big data and CRM, influence e-commerce sales across different countries?" We have also incorporated key theoretical perspectives and empirical findings from existing literature to provide a comprehensive backdrop for our study. Our research addresses the unexplored intricacies of the relationship between big data, CRM, and e-commerce sales, shedding light on the nuanced impacts of these technological tools.
Necessity and Timing:
The need to address this issue stems from the rapid digital transformation and the escalating importance of data-driven decision-making in the contemporary business landscape. Our research is timely, as businesses can remain competitive by integrating technology and data analytics to enhance e-commerce sales, yet lack nuanced insights into the differential impacts of big data and CRM.
Knowledge Gap:
We have identified a knowledge gap in the existing literature, where the intricate relationships between big data, CRM, and e-commerce sales are often oversimplified or addressed in isolation. Our research provides a holistic view, unveiling the contrasting impacts of these technological tools and offering fresh, empirical insights that challenge and enrich prevailing notions.
Theoretical Contribution:
Our theoretical contribution lies in the nuanced insights we offer into the symbiotic yet complex relationship between technology and e-commerce. We have unveiled the dichotomous impact of CRM and affirmed the significant positive influence of big data, adding depth and complexity to the existing body of knowledge.
Practical Implications:
Our research highlights the nuanced impact of CRM tools. While CRM's role in marke-ting didn't show a significant impact on e-commerce sales, its application elsewhere seemed to have a counterintuitive effect. This insight is invaluable for businesses that might be considering a blanket application of CRM tools. Instead of indiscriminately deploying CRM across all functions, businesses should be more strategic, focusing on areas where CRM can genuinely add value. Our findings are particularly relevant for e-commerce businesses, digital marke-ting professionals, and decision-makers in organizations that are at the crossroads of technology adoption. For instance, the positive correlation we identified between big data usage and e-commerce sales underscores the strategic value of data-driven deci-sion-making. Businesses can leverage this insight by investing more in big data analy-tics tools and training, ensuring that their strategies are informed by real-time data and insights.
Addressing the Research Question:
Our findings, derived from a meticulous analysis of Eurostat data, directly address our research question. We have unveiled the significant positive impact of big data and the nuanced, dichotomous effects of CRM on e-commerce sales, offering businesses strategic insights for technology adoption.
Surprising Insights:
Our research could surprise a knowledgeable reader by unveiling the contrasting impacts of CRM in marketing versus other functions, a nuance often overlooked in existing literature. We have provided empirical evidence that enrichs the academic discourse and offers businesses nuanced insights for optimizing CRM strategies.
Timeliness of Recommendations:
Our recommendations are timely, offering businesses real-time insights to navigate the current digital transformation landscape. The strategic insights derived from our research are pivotal for businesses looking to optimize their technology adoption strategies amidst the rapidly evolving digital commerce ecosystem.
Limitations:
We have acknowledged the limitations of our study, including the reliance on Eurostat data and the potential for oversimplification of variables. These limitations offer avenues for future research, inviting a more granular, diversified approach to explore the intricate dynamics of technology and e-commerce.
Expanded Conclusion:
The conclusion has been expanded to offer a comprehensive summation of our findings, theoretical contributions, practical implications, and the addressed knowledge gap.
Please see the attachment. We also highlighted the added text.

Round 2
Reviewer 3 Report
All my comments are addressed.